# Clinical features, antimicrobial susceptibility patterns and genomics of bacteria causing neonatal sepsis in a children's hospital in Vietnam: protocol for a prospective observational study

Nguyen Duc Toan,[1,2,3,4] Thomas C Darton,[1,5] Christine J Boinett,[1] James I Campbell,[1] Abhilasha Karkey,[1] Evelyne Kestelyn,[1,2] Le Quoc Thinh,[3] Nguyen Kien Mau,[3] Pham Thi Thanh Tam,[3] Le Nguyen Thanh Nhan,[1,3] Ngo Ngoc Quang Minh,[3] Cam Ngoc Phuong,[6] Nguyen Thanh Hung,[3,4] Ngo Minh Xuan,[4] Tang Chi Thuong,[4,7] Stephen Baker[1,2,8]

▸ Prepublication history for this paper is available online. To view these files, please visit the journal online (http://dx.doi.org/10.1136/bmjopen-2017-019611).

For numbered affiliations see end of article.

**Correspondence to**
Dr Stephen Baker;
sbaker@oucru.org

## ABSTRACT

**Introduction** The clinical syndrome of neonatal sepsis, comprising signs of infection, septic shock and organ dysfunction in infants ≤4 weeks of age, is a frequent sequel to bloodstream infection and mandates urgent antimicrobial therapy. Bacterial characterisation and antimicrobial susceptibility testing is vital for ensuring appropriate therapy, as high rates of antimicrobial resistance (AMR), especially in low-income and middle-income countries, may adversely affect outcome. Ho Chi Minh City (HCMC) in Vietnam is a rapidly expanding city in Southeast Asia with a current population of almost 8 million. There are limited contemporary data on the causes of neonatal sepsis in Vietnam, and we hypothesise that the emergence of multidrug resistant bacteria is an increasing problem for the appropriate management of sepsis cases. In this study, we aim to investigate the major causes of neonatal sepsis and assess disease outcomes by clinical features, antimicrobial susceptibility profiles and genome composition.

**Method and analysis** We will conduct a prospective observational study to characterise the clinical and microbiological features of neonatal sepsis in a major children's hospital in HCMC. All bacteria isolated from blood subjected to whole genome sequencing. We will compare clinical variables and outcomes between different bacterial species, genome composition and AMR gene content. AMR gene content will be assessed and stratified by species, years and contributing hospital departments. Genome sequences will be analysed to investigate phylogenetic relationships.

**Ethics and dissemination** The study will be conducted in accordance with the principles of the Declaration of Helsinki and the International Council on Harmonization Guidelines for Good Clinical Practice. Ethics approval has been provided by the Oxford Tropical Research Ethics Committee 35-16 and Vietnam Children's Hospital 1 Ethics Committee 73/GCN/BVND1. The findings will be disseminated at international conferences and peer-reviewed journals.

### Strengths and limitations of this study

▸ Little is known about the current aetiological agents of neonatal sepsis in Vietnam. This prospective study will integrate clinical assessments with microbiological and detailed whole genome sequence data to characterise the aetiology and outcome of neonatal sepsis in this high-mortality setting.
▸ This study is being performed at the largest secondary/tertiary paediatrics centre in Southern Vietnam. Data collection at a single site may limit the applicability to other hospitals in the country.
▸ Other limitations encountered in designing this study include ethical issues involved in collecting samples from severely ill neonates, lack of current transferable definitions for neonatal sepsis phenotypes and stochastic variations in numbers of cases recruited due to seasonal variation and continuous changes in community antimicrobial and vaccine use.
▸ Some contamination of blood cultures is unavoidable in our setting and therefore accurately classifying some isolates as true pathogens in certain cases may be challenging.

**Trial registration number** ISRCTN69124914; Pre-results.

## BACKGROUND

Neonatal sepsis is widely recognised as a clinical syndrome of systemic inflammation in response to, or occurring the same time as, a possible or proven infection (frequently by identifying bacterial bloodstream infection) occurring in children ≤28 days of age. A consensus definition of neonatal sepsis has remained a challenge.[1] Globally, the incidence

of neonatal sepsis is estimated to be 1–5 cases/1000 live births but is lower in full-term neonates (1–2 cases/1000 live births), in whom the incidence is higher in men than women.[2 3] Early-onset sepsis is defined as the start of sepsis symptoms within 72 hours of birth[1 4] and is often caused by vertical transmission of pathogens during delivery as a result of chorioamnionitis or maternal genital tract colonisation.[5] Late-onset sepsis, occurring after 72 hours from birth,[1 6] may be caused by similar vertical transmission or horizontal transmission mechanisms due to direct contact with the surrounding environment, attendant healthcare staff or any invasive procedures.[7]

## Neonatal sepsis in Southeast Asia

Neonatal sepsis remains a leading cause of neonatal hospital admission, morbidity and mortality in low-income and middle-income countries (LMICs).[8] In this setting, bacterial infection, including bacteraemia, is complicated by multidrug resistance, particularly related to healthcare acquired infection, and effective management of neonatal sepsis is increasingly problematic.[8] Recently, WHO has acknowledged the problem of antimicrobial resistance (AMR) as an endemic and widespread problem in LMICs.[9] In many LMICs untreatable bacterial infections with broadly AMR pathogens are no longer a threat but a common reality. AMR in LMICs represents one of the biggest threats to global health and are one of the greatest current challenges in infectious disease research.

While AMR is an issue with all types of bacterial infection, the issue is most acute in management of clinical sepsis. This is a particular problem in neonates due to high mortality/morbidity rates and the timely need for rapid detection and treatment of the causative pathogen. Sepsis demonstrates extensive geographical diversity in both aetiology and proportions of AMR bacteria isolated.[10 11] Understanding the local and regional epidemiology of sepsis in hospitalised neonates is crucial in the development of rational management and treatment guidelines, especially in high-risk AMR LMICs locations like Vietnam.

## Sepsis and AMR in Vietnam

Bacterial sepsis is classified into two major groups according to place of acquisition. Hospital-acquired sepsis is defined in patients with clinical manifestations of sepsis and a confirmatory blood culture collected >48 hours following hospital admission.[12] Hospital-acquired sepsis is a major threat to patient safety, and in locations with poor surveillance and infection control programme such infections are associated with high mortality rates. The incidence of AMR bloodstream infections in Vietnam has increased over recent years and is predicted to increase further.[13] This trend has comprised an increase in both Gram-negative and Gram-positive pathogens, chiefly *Escherichia coli*, *Klebsiella pneumoniae*, *Acinetobacter baumannii*, *Staphylococcus aureus* and enterococci.

Community-acquired sepsis is also an important cause of fever among patients admitted to hospitals across South and Southeast Asia. In our practice, community-acquired

sepsis is defined as the presence of sepsis with a confirmatory blood culture collected within 48 hours after hospital admission.[14] Distinguishing bacterial infections from other common causes of fever, such as malaria or dengue, can be challenging without diagnostic laboratory support.[15] Important causes of community-acquired bloodstream infection (CA-BSI) in LMICs include *Salmonella* serovars Typhi and Paratyphi A, *E. coli*, *K. pneumoniae*, *S. aureus* and *Streptococcus pneumoniae*. We have reported increasing levels of AMR in these common community-acquired pathogens and highlighted the difficulties of accurate diagnosis with traditionally available diagnostics. For example in Nepal and Vietnam antimicrobial resistance in *Salmonella* Typhi and *Salmonella* Paratyphi A has severely restricted the options available for antimicrobial treatment.[16]

## Changing aetiology of BSI in Vietnam

The aetiology of CA-BSI in Vietnam has changed considerably over the 20 years. A previous study documented the decline of *Salmonella* Typhi from 2002, the predominant pathogen until this point and the subsequent increase in non-typhoidal *Salmonella* and other opportunistic HIV-associated pathogens.[17] This shift is likely to reflect a changing landscape of infectious disease related to the HIV epidemic, urbanisation and secondary social determinants within Vietnam. Vietnam, as with many countries in Asia, is undergoing a rapid economic transition, and programmes to improve sanitary conditions have reduced the overall risk of waterborne infections. HIV-associated opportunistic pathogens have now emerged as the leading cause of bloodstream infections and the primary cause of mortality in hospitalised adult patients in this location. These studies were performed at the Hospital for Tropical diseases in Ho Chi Minh City and thus included mainly adults, including those with HIV, and children but not neonates.[18–20] Therefore, these observations may not be fully representative of the situation in neonates.

## Knowledge gaps

Little is known about contemporary antimicrobial susceptibility patterns and their underlying genetic determinants in the major causes of neonatal bacterial of sepsis in Vietnam. Furthermore, the impact of antimicrobial susceptibility and other virulence factors on disease progression and outcome in neonates in LMICs is also not well documented. To address these issues, we aim to investigate the aetiology of pathogens associated with bacterial sepsis in neonates and to detail the effects of reduced antimicrobial susceptibility on the outcome of sepsis in neonates. This will be a clinical and microbiology laboratory research project between Children's Hospital 1 and the Oxford University Clinical Research Unit (OUCRU) in Vietnam. This study will be a conduit for introducing molecular biology for bacteriology into routine hospital care at this children's hospital and will lead to future studies investigating appropriate empirical treatment for bacteraemia and the impact of antimicrobial

resistance on the outcome of sepsis in the paediatric and neonatal population in Vietnam.

## Rationale, aim and objectives

To understand the causes of neonatal sepsis and to best inform antimicrobial treatment regimes in our setting, we will perform a prospective observational study at Children's Hospital 1 in Vietnam from 2017 to 2019. There are limited contemporary data on the causes of bacterial sepsis in neonates in Vietnam. We hypothesise that there have been recent increases in multidrug resistant Gram-negative bacteria causing sepsis in this high-risk group and that methicillin-resistant *S. aureus* has emerged as an important pathogen. We further aim to investigate the clinical features, major causes of neonatal sepsis and the distribution of pathogens by departments, their antimicrobial susceptibility patterns and the genomic profiles of the isolated bacteria as well as their association with disease outcomes.

### Primary objectives

► To describe the clinical characteristics of neonates with sepsis, including community and hospital-acquired sepsis, early-onset and late-onset sepsis.
► To determine the aetiology of neonatal sepsis and the distribution of pathogens by clinical departments including the Neonatology Department and the Neonatal Intensive Care Unit.
► To determine the antimicrobial susceptibility profiles of the bacteria causing neonatal sepsis and the AMR profiles occurring in community and hospital-acquired infections.
► To analyse the impact of specific bacteria and AMR profile on the outcomes (mortality, length of stay and cost of treatment) of neonates with sepsis.
► To determine the genome sequences of bacterial strains associated with neonatal sepsis.

### Secondary objectives

► To determine the AMR profiles and gene distribution of isolated bacteria by clinical departments to add insight into the circulation of bacteria associated with hospital acquired infections.
► To study the genes catalysing resistance to antimicrobials commonly used to treat neonatal sepsis (specifically third/fourth generation cephalosporins, fluoroquinolones and carbapenems).

## METHODS

### Study design

This protocol describes a prospective, non-interventional, observational study to characterise the clinical features of neonates with sepsis at Children's Hospital 1 in Ho Chi Minh City in Vietnam between 2017 and 2019, the microbial population structure, antimicrobial susceptibility patterns and the AMR genes of the bacteria causing that sepsis. All organisms isolated from blood will be stored and archived for molecular characterisation.

### Study site

Children's Hospital 1 (Neonatology Department, Neonatal Intensive Care Unit, Microbiology Department) is in Ho Chi Minh City in Vietnam. The estimated population of the city was 8.4 million in 2016, and 23.8% are children 0–14 years of age.[21] Children's Hospital 1 is the largest tertiary paediatrics centre in Southern Vietnam with 1400 inpatient beds and >1600 staff members. The hospital receives ~1.5 million outpatient visits and 95 000 admissions each year. Care is provided to all children <15 years old from Ho Chi Minh City and other provinces of Southern Vietnam. The neonatal centre at this hospital currently has 120 inpatient beds for the neonatology department and additional 30 beds in the neonatal intensive care unit. The overall mean rate of positive blood cultures in our hospital is 7% per year.

## DEFINITIONS

### Definition of sepsis

A sepsis episode in this study is defined as isolation of a clinically relevant pathogen from ≥1 blood culture, drawn from a neonate with ≥1 clinical or laboratory sign of sepsis (table 1).[22]

### Diagnosis of neonatal sepsis

Systematic guidelines concerning which patients should have blood cultures performed are not strictly defined in our hospital, although blood culture results are used to confirm the diagnosis of sepsis in neonates with a compatible clinical presentation. We use the criteria suggested by the European Medicines Agency in 2010 for the diagnosis of 'probable sepsis' and 'confirmed sepsis' in neonates[22]:
► Probable sepsis: ≥2 clinical and ≥2 laboratory signs;
► Confirmed sepsis: ≥1 positive culture of a pathogen and ≥1 clinical or laboratory sign.

## SAMPLE SIZE

In this prospective observational study, we aim to recruit all patients with available data who fulfil the inclusion criteria and are admitted to Children's Hospital 1 in Ho Chi Minh City in Vietnam from 2017 to 2019. Based on retrospective surveillance data, we estimate recruitment of 800 participants during the study period. Blood cultures will be performed in all cases, we expect to yield ~400 bacterial isolates.

### Participant selection and recruitment

#### Inclusion criteria

Neonates (≤1 month of age) with a diagnosis of 'probable' or 'confirmed' sepsis who have had a blood culture taken and who are an inpatient at Children's Hospital 1 will be recruited into the study after written informed consent has been given by a parent or guardian.

### Exclusion criteria

Patients will be excluded when informed consent is not provided, the length of hospital stay less than 24 hours, imminent and inevitable death or the patient has been

| Table 1 | Clinical and laboratory signs of neonatal sepsis |
|---|---|
| Clinical signs of sepsis | ▶ Abnormal body temperature (core temperature >38.5°C or <36°C and/or temperature instability); <br> ▶ Cardiovascular instability (bradycardia (mean heart rate <10th percentile for age in the absence of external vagal stimulus, beta blockers or congenital heart disease or otherwise unexplained persistent depression over a 0.5–4 hour time period) or tachycardia (mean heart rate >2 SD above normal for age in the absence of external stimulus, chronic unexplained persistent elevation over a 0.5–4 hour time period) and/or rhythm instability, reduced urinary output (<1 mL/kg/hour), hypotension (mean arterial pressure <5th percentile for age), mottled skin, impaired peripheral perfusion); <br> ▶ Respiratory instability (apnoea episodes or tachypnoea episodes (mean respiratory rate >2 SD above normal for age) or increased oxygen requirements or requirement for ventilator support); <br> ▶ Gastrointestinal (feeding intolerance, poor sucking, abdominal distension); <br> ▶ Skin and subcutaneous lesions (petechial rash, sclerema); <br> ▶ Non-specific (irritability, lethargy, hypotonia). |
| Laboratory signs of sepsis | ▶ White cells $<4\times10^9$ cells/L or $>20\times10^9$ cells/L <br> ▶ Immature to total neutrophil ratio (I/T) >0.2 <br> ▶ Platelet count $<100\times10^9$/L <br> ▶ C reactive protein >15 mg/L or procalcitonin ≥2 ng/mL <br> ▶ Glucose intolerance (hyperglycaemia (blood glucose >180 mg/dL or 10 mmol/L) or hypoglycaemia (blood glucose <45 mg/dL or 2.5 mmol/L)) <br> ▶ Metabolic acidosis (base excess <−10 mEq/L or serum lactate >2 mmol/L) |

previously recruited in the study. Investigators will review all of the mortality records during the time period of the study to try and identify how many of these cases may have been missed and whether there were any common characteristics in these participants.

### Identification of participants

All doctors and nurses in the Department of Neonatology and Neonatal Intensive Care Unit of the study hospital will be informed about, and trained for, this clinical investigation. In addition, those working in the Department of Neonatology and Neonatal Intensive Care Unit will also be involved in the study. These staff will be trained to identify eligible patients and how to notify investigators.

### Informed consent

Trained, Good Clinical Practice (GCP) accredited, members of the study team will collect informed consent. The team will discuss the study with the accompanying parent/guardian, or, if both parents are deceased or not actively involved in child care, the main long-term carer of the child will be accepted as the guardian and considered able to give consent for the study. Study staff will describe the purpose of the study, the study procedures, possible risks/benefits, the rights and responsibilities of participants and alternatives to enrolment. The parent/guardian will be invited to ask questions, which will be addressed by study staff, and they will be provided with appropriate contact numbers if they have any subsequent questions. If the parent/guardian agrees for the child to participate, they will be asked to sign and date an informed consent form. A copy of the patient information sheet and the informed consent form will be given to them to keep. In addition to the procedures above, illiterate signatories will have the informed consent form read to them in the presence of a

witness who will sign to confirm this. The parent/guardian can withdraw from the study at any time (verbally) without affecting the care that the child will receive. If the parent/guardian decides at any time to take the child out of the study, no new information will be collected. However, information collected on the child up until that point will still be used. All patient information sheets and consent forms will be written in the local language and will use terms that are easily understandable.

### Study procedures

An investigator will routinely record and collect demographic, clinical and laboratory information of the patients, the date of blood draw, the number of blood culture bottles inoculated, the result of the culture (whether positive or negative) and the susceptibility of the isolate to commonly used antimicrobials. Data from these records will be subsequently entered into CliRes Data Management System of OUCRU. These will be source data for this study. The number of patients admitted to the hospital annually will be obtained from hospital records. As part of this study, we request that all isolates from blood are stored and archived at −80°C. These isolates will be recultured and the identification will be reconfirmed. Selected isolated organisms from blood will have further molecular characterisation at a later date.

## DATA COLLECTION
### Demographic and clinical assessments

Data on neonatal sepsis at Children's Hospital 1 will be collected to the case report form. These data will include administrative data, demographic data, clinical characteristics, laboratory results, diagnoses, treatments and outcomes.

The Neonatal Therapeutic Intervention Scoring System will be used to estimate the disease severity.[23]

## Microbiological assessments

Available routine microbiology data on neonatal bloodstream infections Children's Hospital 1. These data will include pathogenic agents isolated from blood culture and antimicrobial susceptibility profile of isolated bacteria (routine panel of antimicrobials). Selected isolates will have additional antimicrobial susceptibility testing, molecular analysis for antimicrobial resistance genes and genome sequencing of selected strains defined by antimicrobial susceptibility data.

## LABORATORY METHODS

### Microbiology testing

When required for checking or to non-routine antimicrobials, antimicrobial susceptibility testing of the pathogens isolated will be performed by disk diffusion using guidelines established by the Clinical and Laboratory Standards Institute and, when required, by minimum inhibitory concentration estimation using the VITEK 2 COMPACT automated machine. Antimicrobial susceptibilities tested will include nalidixic acid, ciprofloxacin, ceftriaxone, cefepime, ampicillin, trimethoprim–sulfamethoxazole, azithromycin, imipenem, colistin and amikacin for all Gram-negative organisms and oxacillin and vancomycin in Gram-positive organisms. The production of extended-spectrum beta lactamases (ESBL) will be investigated using the double-disc synergy test by comparing zone sizes between ceftazidime discs against ceftazidime–clavulanic acid discs and cefotaxime discs against cefotaxime–clavulanic acid discs. Isolates with an increase in diameter of inhibitory zone of equal to or more than 5 mm by the synergy of clavulanate will be considered ESBL positive.

Organisms including coryneforms (*Corynebacterium*, etc), *Micrococci*, *Propionibacterium*, *Bacillus*, alpha haemolytic *Streptococci*, environmental Gram-negative bacilli and non-pathogenic *Neisseria* will be considered potential contaminants. The pathogen-contaminant decision will be made based on the clinical relevance of the isolated bacteria and the independent assessments by two qualified medical microbiologists. If there is disagreement, then the case will be discussed until a decision is reached.

### Bacterial storage

Organisms will be subcultured onto 5% blood agar and the purity of the isolate will be tested before storage in 20% glycerol at −80°C.

### Isolation of nucleic acids

Isolates will be recultured and their identification rechecked. DNA will be extracted from bacterial isolates using the Wizard Genomic DNA Extraction Kit (Promega, Fitchburg, Wisconsin, USA). The quality and concentration of the DNA will be assessed using a nanodrop spectrophotometer prior to PCR amplification and the Quant-IT Kit (Invitrogen, Carlsbad, California, USA) prior to DNA sequencing.

### PCR for resistance genes

The primary focus study is to investigate the distribution of antimicrobial resistance genes in bacteria causing neonatal sepsis. Therefore, all Gram-negative organisms will be investigated by PCR to detect genes catalysing resistance to cephalosporins, fluoroquinolones and carbapenems. Conventional PCR will be performed for the following classes of resistance genes using previously described methods. The multiplex and monoplex PCRs are described in these publications. PCR will be used to detect AmpC, ESBL (including CTX-M15), NDM, qnr, OXA, KPC and mecA/Van.[11 24–29]

### Genome sequencing

Selected organisms (on the basis of their susceptibility profiles and resistance gene content) will be genome sequenced. We aim to sequence the greatest cross-section of organism groups as possible (ie, all *Staphylococci* or all *Klebsiella*). Selected bacterial isolates will be sequenced at OUCRU in Vietnam. Briefly, index-tagged paired end Illumina sequencing libraries will be prepared using one of 96 unique indexing tags as previously described. These will be combined into pools of uniquely tagged libraries and sequenced on the Illumina Genome Analyzer or HiSeq sequencer according to manufacturer's protocols to generate tagged 54–100 bp paired-end reads. This is previously approached for describing Gram-negative organisms and *Staphylococcus*.[20 30 31]

## ANALYSIS PLAN

### Statistical comparisons

Data will be presented in the form of tables and bar charts for descriptive variables, that is, number of organisms per year and number of AMR organisms per year. Statistical comparisons of features between groups (positive/negative blood culture, Gram-negative/Gram-positive bacteria, survival/non-survival, etc) and time trend analysis of the cultured isolates by month and the antimicrobial susceptibility patterns will be conducted. These data will be placed in the context of the broader population by comparison of these data with historical laboratory records of pathogens isolated from patients with bloodstream infections. Historical data from both neonates and older children will be analysed descriptively, and where appropriate, time trend analyses will be performed to determine significant alterations in bloodstream infection aetiology. All statistical analysis will be performed using Stata V.14 and R. P values of ≤0.05 will be considered significant.

### Antimicrobial resistance genes and genome sequencing

The presence/absence of antimicrobial resistance genes will be reported as proportions per organism and then

stratified by organism, year and hospital department. Genome sequences will be determined to study phylogenetic relationships, the presence/absence of virulence genes and also AMR gene content and first analysed by species and then group by their Gram-stain results. Briefly, for phylogenetic analysis, chromosomal single nucleotide polymorphism (SNP) alleles will be concatenated for each strain to generate a multiple alignment of all SNPs. For maximum likelihood (ML) analysis, RAxML will be run using the generalised time-reversible model and 1000 bootstrap pseudoreplicate analyses were performed to assess support for the ML phylogeny. Root-to-tip branches will be extracted from the ML tree using the programme TreeStat. The relationship between root-to-tip distances and year of isolation will be analysed using linear regression. For Bayesian Evolutionary Analysis Sampling Trees (BEAST) analysis (V1.6), a GTR+Γ substitution model and defined tip dates as the date of isolation will be used.[30 31] To detect the presence or absence of genes read sets will be assembled using the de novo short read assembler Velvet and Velvet Optimizer. Organism specific read sets will then be aligned to the pangenome. Taxonomic investigation of accessory and AMR genes will be performed using MG-RAST V.3.2.

### Ethics, regulatory approvals and governance
This study is sponsored by the University of Oxford and will be monitored by the Clinical Trials Unit at OUCRU. The Principal investigator (SB) will ensure that this study is conducted in accordance with the principles of the Declaration of Helsinki and the terms of approval of the appropriate ethical committees.[32] The study will be conducted in full conformity with relevant regulations and with the International Council on Harmonisation Guidelines for GCP.[33] This protocol and the relevant supporting document have already had the approvals of the Oxford Tropical Research Ethics Committee and the institutional review board of Children's Hospital 1. The investigators will submit and, where necessary, obtain approval from the above parties for all substantial amendments to the original approved documents.

### Dissemination and public engagement
Data from this study will be of interest to the scientific and clinical research communities. An informative resource for managing sepsis will be made available to local clinicians, clinical microbiologists and infection control policy developers. Study data will be reported according to the Strengthening the Reporting of Observational studies in Epidemiology guidance for reporting observational studies.[34] The authors (and their respective positions in the author list) will be agreed prior to the start of the study in accordance with the guidelines of the International Committee of Medical Journal Editors. In line with Wellcome Trust policy that the results of publicly funded research should be freely available, manuscripts arising from this study will be submitted to peer-reviewed journals which enable Open Access. In line with research transparency and greater access to data sharing policy of OUCRU in Vietnam will be implemented. This policy is based on a controlled access approach with a restriction on data release that would compromise an ongoing trial or study. Data exchange complies with Information Governance and Data Security Policies in all of the relevant countries.

## DISCUSSION
As a conduit for introducing molecular biology for bacteriology into routine hospital care in Vietnam, the

| Table 2 | Study summary |
|---|---|
| Title | The clinical features, antimicrobial susceptibility patterns and genomics of bacteria causing neonatal sepsis in a children's hospital in Vietnam |
| Design | Observational prospective study<br>All organisms isolated from blood will be stored and archived for molecular characterisation |
| Participants | Neonates (≤1 month of age) with sepsis |
| Planned enrolment period | 2017–2019 |
| Primary objectives | ► To investigate the clinical characteristics of neonatal sepsis;<br>► To define the aetiology, the percentage of positive blood culture and major causes of sepsis;<br>► To investigate the antimicrobial susceptibilities of the pathogens causing sepsis and the rate of antimicrobial resistance;<br>► To measure the impact of sepsis on the severity of disease and the outcomes (mortality rate, length of stay and cost of treatment) of hospitalised neonates;<br>► To analyse the genome sequences of bacterial strains causing neonatal sepsis. |
| Secondary objectives | ► To analyse the antimicrobial resistance profiles and gene distribution by clinical departments to add insight into the circulation of bacteria causing nosocomial infections;<br>► To study the genes catalysing resistance to the antimicrobials commonly used to treat neonatal sepsis (specifically third-generation/fourth-generation cephalosporins, fluoroquinolones and carbapenems). |

study is unique and is planned to lead to future studies investigating appropriate empirical treatment for bacteraemia and the impact of AMR on the outcome of sepsis in the neonatal and paediatric population in Vietnam. By studying and defining disease aetiology, antimicrobial susceptibility patterns and disease outcome, we plan to develop an improved approach to managing bloodstream infections in our setting, and we will use these data to initiate intervention studies focused on preventing sepsis with AMR pathogens in neonates. Table 2 shows the summary of this study.

## Duration and current status of study

The first patient was recruited in January 2017. At the current time, the recruitment is ongoing. The expected end date for recruitment is 31 December 2019. We expect to have completed our data analysis plan with a view of results by June 2020.

**Author affiliations**
[1]Hospital for Tropical Diseases, Wellcome Trust Major Overseas Programme, Oxford University Clinical Research Unit, Ho Chi Minh City, Vietnam
[2]Centre for Tropical Medicine and Global Health, Nuffield Department of Clinical Medicine, University of Oxford, Oxford, UK
[3]Intensive care department, Children's Hospital, Ho Chi Minh City, Vietnam
[4]General planning, Pham Ngoc Thach University of Medicine, Ho Chi Minh City, Vietnam
[5]Department of Infection, Immunity and Cardiovascular Disease, University of Sheffield Medical School, Sheffield, UK
[6]Hanh Phuc International Hospital, Binh Duong, Vietnam
[7]Department of Health, Ho Chi Minh City, Vietnam
[8]Department of Medicine, University of Cambridge, Cambridge, UK

**Contributors** All authors contributed to the conception and design of the study. NDT, TCD and SB drafted the protocol of the study. All authors read and critically revised the protocol and this article prior to submission.

**Funding** This study is supported by the Royal Society and the Wellcome Trust (grant 100087/Z/12/Z). The Oxford University Clinical Research Unit is a Major Overseas Programme funded by the Wellcome Trust (grant 089276/2/09/2). TCD is supported by the National Institutes of Health Research (grant 3557) and supported by a Clinical Lecturer Starter Grant from the Academy of Medical Sciences and Wellcome Trust (grant SGCL015/1005).

**Competing interests** None declared.

**Patient consent** Parental/guardian consent obtained.

**Ethics approval** Oxford (Oxford Tropical Research Ethics Committee 35-16) and Vietnam ().

**Provenance and peer review** Not commissioned; externally peer reviewed.

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
