## [Reviewer comments · BMJ Open]

ARTICLE DETAILS

TITLE (PROVISIONAL)	The clinical features, antimicrobial susceptibility patterns, and genomics of bacteria causing neonatal sepsis in a children's hospital in Vietnam: protocol for a prospective observational study
AUTHORS	Toan, Nguyen; Darton, Thomas; Boinett, Christine; Campbell, James; Karkey, Abhilasha; Kestelyn, Evelyne; Thinh, Le; Mau, Nguyen; Tam, Pham Thi Thanh; Nhan, Le Nguyen Thanh; Minh, Ngo Ngoc Quang; Phuong, Cam; Hung, Nguyen Thanh; Xuan, Ngo; Thuong, Tang; Baker, Stephen

VERSION 1 – REVIEW

REVIEWER	Nazima Pathan University of Cambridge, United Kingdom
REVIEW RETURNED	19-Sep-2017

GENERAL COMMENTS	An important study examining the problem of antimicrobial resistance in a LMIC community in Vietnam. The protocol is clear, focused and easy to follow. The objectives are appropriate and the study rationale is well described and justified. The authors have put together an excellent research plan and I look forward to following it's progress.
--

REVIEWER	Pascal Lavoie BC Children's Hospital Research Institute, Canada
REVIEW RETURNED	08-Oct-2017

GENERAL COMMENTS	The study is important, well written and well designed. Data from this study will likely be useful to the scientific and local community. Two limitations that are not discussed are the possibility that the lack of consent in a large proportion of the subject, or in subjects who die within 24h due to a severe sepsis may be missed as per the statement of exclusion criteria. Another limitation is due to the possibility that a number of positive blood culture might be due to contamination at the time of sampling. How will the authors assess this possibility? I would suggest that the authors attach their case-report form as supplemental material, with clear definitions, which will help whoever may conduct similar studies. This will also help review whether the study objectives are likely to be met or not.
--

	It is not clear how eligible subjects will be screened/identified, by who and what will happen/which data will be kept in cases subjects do not consent. Following that last point, there may be value in collecting a minimal set of completely de-identified data (e.g. population denominator, microbial blood culture results, antibiotic resistance pattern) provided that it is acceptable to the local community and to the author's institutional ethics board.
--	---

REVIEWER	Raymond Josette Hôpital Cochin, Bacteriology, France
REVIEW RETURNED	10-Oct-2017

GENERAL COMMENTS	The aim of this study is to specify the epidemiology of early and late neonatal infections in Vietnam. This study is necessary. However, some details need to be clarified.  - Precise the 2 laboratory signs to define a probable sepsis. Since the study is based on the isolation of 1 bacteria, only the confirmed sepsis must be taken into account  - Precise bacteria that will be considered contaminants - The authors estimated 800 participants with 400 bacterial isolates? - That is to say one hemocult on two positive? This is not realistic - PCR for resistance genes: Research of CTXM-15, OXA-48 and KPC are lacking. - Genome sequencing: This is the most important point. What is the purpose of this sequencing? What is the exact question asked? To which strains will the strains of this study be compared? To those of the environment? Caregivers? It cannot only be a descriptive study.
--

VERSION 1 – AUTHOR RESPONSE

Comment:

Two limitations that are not discussed are the possibility that the lack of consent in a large proportion of the subject, or in subjects who die within 24h due to a severe sepsis may be missed as per the statement of exclusion criteria.

Response:

A lack of consent of eligible participants could happen but in a very small proportion of the research population. The parents/guardians of neonates are informed that there is no risk in participating in this study because it does not involve in testing any new drugs or interventions and we also aim to find the cause of the sepsis so that the patients receive the correct medication. These have been added to the strengths and limitations

The lack of consent of eligible participants or in subjects who die within 24 hours due to severe sepsis may happen. The investigators will review all of the mortality records during the time period of the study to try and identify how many of these cases may have been missed and whether there were any common characteristics in these participants.

We have added this point to the section 'Exclusion criteria'.

Comment:

Another limitation is due to the possibility that a number of positive blood culture might be due to contamination at the time of sampling. How will the authors assess this possibility?

Response:

We admit that the contamination of blood cultures is unavoidable. In this study, the pathogen-contaminant decision will be made based on the clinical relevance of the isolated bacteria and the independent assessments by two qualified medical microbiologists. We have added this point to the section 'Microbiology testing'.

Comment: I would suggest that the authors attach their case-report form as supplemental material, with clear definitions, which will help whoever may conduct similar studies. This will also help review whether the study objectives are likely to be met or not.

Response:

Our case-report form has been attached as a supplemental material.

Comment: It is not clear how eligible subjects will be screened/identified, by who and what will happen/which data will be kept in cases subjects do not consent.

Response:

Neonates with a diagnosis of "probable" or "confirmed" sepsis who have had a blood culture taken and who are an in-patient at Children's Hospital 1 will be recruited into the study, after written informed consent has been given by a parent or guardian. Patients will be excluded when informed consent is not provided, the length of hospital stay less than 24 hours, imminent and inevitable death, or the patient has been previously recruited in the study. All doctors and nurses in the Department of Neonatology and Neonatal Intensive Care Unit of the study hospital will be informed about, and trained for, this clinical investigation. In addition, those working in the Department of Neonatology, and Neonatal Intensive Care Unit will also be involved in the study. These staff will be trained to identify eligible patients and how to notify investigators. These have already been mentioned in the section 'Participant selection and recruitment'.

The parent/guardian can withdraw from the study at any time (verbally) without affecting the care that the child will receive. If the parent/guardian decides at any time to take the child out of the study, no new information will be collected. However, information collected on the child up until that point will still be used. All of these points have been mentioned in the section 'Informed consent'.

Comment: Following that last point, there may be value in collecting a minimal set of completely de-identified data (e.g. population denominator, microbial blood culture results, antibiotic resistance pattern) provided that it is acceptable to the local community and to the author's institutional ethics board.

Response:

These data will be placed in the context of the broader population, by comparison of these data with historical laboratory records of pathogens isolated from patients with blood stream infections. Historical data from both neonates and older children will be analysed descriptively, and where appropriate, time trend analyses will be performed to determine significant alterations in bloodstream infection aetiology.

This has been added to the section "Analysis plan - Statistical comparisons".

Comment: Precise the 2 laboratory signs to define a probable sepsis. Since the study is based on the isolation of 1 bacteria, only the confirmed sepsis must be taken into account

Response:

Laboratory signs are described in Table 1 and include white blood cells $<4 \times 10^9$ cells/L or $>20 \times 10^9$ cells/L; immature to total neutrophil ratio (I/T) >0.2 ; platelet count $<100 \times 10^9$ /L; CRP >15 mg/L or procalcitonin ≥ 2 ng/mL; glucose intolerance (hyperglycemia [blood glucose >180 mg/dl or 10 mmol/L] or hypoglycemia [blood glucose <45 mg/dl or 2.5 mmol/L]); metabolic acidosis (base excess <-10 mEq/L or serum lactate >2 mmol/L)

Both probable sepsis and confirmed sepsis will all be included in this study. Doctors and nurses working in clinical wards will identify eligible patients and notify investigators about “probable sepsis” cases; the patients will then be recruited. When we have the blood culture results, we will know that whether the confirmation of sepsis (“confirmed sepsis”) in those cases is made or not.

Comment: Precise bacteria that will be considered contaminants

Response:

Organisms including Coryneforms (Corynebacterium, etc.), Micrococci, Propionibacterium, Bacillus, alpha haemolytic Streptococci, environmental Gram-negative bacilli, and non-pathogenic Neisseria will be considered potential contaminants.

In this study, the pathogen-contaminant decision will be made based on the clinical relevance of the isolated bacteria and the independent assessments by two qualified medical microbiologists. If there is disagreement then the case should be discussed until a decision is reached. We have added this to the section ‘Microbiology testing’.

Comment: The authors estimated 800 participants with 400 bacterial isolates? - That is to say one hemocult on two positive? This is not realistic

Response:

We think the original estimation is fine – we expect to recruit 800 probable/confirmed cases of sepsis, and of whom 50% (i.e. 400) are anticipated to have ≥ 1 positive blood culture. Given the selection of the population and the nature of neonatal sepsis we predict this is realistic

Comment: PCR for resistance genes: Research of CTXM-15, OXA-48 and KPC are lacking.

Response:

These have been added

Comment: Genome sequencing: This is the most important point. What is the purpose of this sequencing? What is the exact question asked? To which strains will the strains of this study be compared? To those of the environment? Caregivers?

Response:

The question asked will be “what is the role of genome content on disease progression and outcome”. Therefore we will sequence all isolates and stratify by species and then assess the role on outcome. Further, we will concatenate all sequence data and investigate gene content on outcome. i.e. investigate cross species resistance gene or siderophores genes that may impact on treatment.

Comment:

It cannot only be a descriptive study.

Response:

This protocol describes a prospective, non-interventional, observational study to characterize the clinical features of neonates with sepsis, the microbial population structure, antimicrobial susceptibility patterns and the AMR genes of the bacteria causing that sepsis. All organisms isolated from blood will be stored and archived for molecular characterization. Statistical comparisons of features between groups (positive/negative blood culture, Gram negative/Gram positive bacteria, survival/non-survival etc.) and time trend analysis of the cultured isolates by month and the antimicrobial susceptibility patterns will be determined.

VERSION 2 – REVIEW

REVIEWER	Pascal Lavoie BC Children's Hospital Research Institute, Vancouver Canada
REVIEW RETURNED	28-Nov-2017

GENERAL COMMENTS	To the question: Is the supplementary reporting complete (e.g. trial registration; funding details; CONSORT, STROBE or PRISMA checklist)? Unfortunately I could not have access to the supplementary files, including the data collection form. This may be due to the online system...
---

REVIEWER	Raymond Josette Hopital cochin
REVIEW RETURNED	17-Nov-2017

GENERAL COMMENTS	The recommended revisions were done This manuscript can now be accepted
--